# Immunosuppressive Treatment in Antiphospholipid Syndrome: Is It Worth It?

**DOI:** 10.3390/biomedicines9020132

**Published:** 2021-02-01

**Authors:** Ilaria Mormile, Francescopaolo Granata, Alessandra Punziano, Amato de Paulis, Francesca Wanda Rossi

**Affiliations:** 1Department of Translational Medical Sciences, University of Naples Federico II, Via S. Pansini 5, 80131 Naples, Italy; ilariamormile@virgilio.it (I.M.); frapagra@hotmail.com (F.G.); alessandra.punziano@unina.it (A.P.); amato.depaulis@unina.it (A.d.P.); 2Center for Basic and Clinical Immunology Research (CISI), WAO Center of Excellence, University of Naples Federico II, 80131 Naples, Italy

**Keywords:** antiphospholipid syndrome, lupus anticoagulant, anticardiolipin, anti-β2-glycoprotein I, catastrophic antiphospholipid syndrome

## Abstract

The antiphospholipid syndrome (APS) is characterized by the development of venous and/or arterial thrombosis and pregnancy morbidity in patients with persistent antiphospholipid antibodies (aPL). Catastrophic antiphospholipid syndrome (CAPS) is a life-threatening form of APS occurring in about 1% of cases. Lifelong anticoagulation with vitamin K antagonists remains the cornerstone of the therapy for thrombotic APS, but frequently the use of anticoagulation may be problematic due to the increased risk of bleeding, drug interactions, or comorbidities. Immunosuppressant drugs are widely used to treat several autoimmune conditions, in which their safety and effectiveness have been largely demonstrated. Similar evidence in the treatment of primary APS is limited to case reports or case series, and studies on a large scale lack. Immunomodulatory drugs may be an emerging tool in managing such particular situations, like refractory obstetrical complications, CAPS, or so-called APS non-criteria manifestations. In addition, immunomodulatory drugs may be useful in patients experiencing recurrent thromboembolic events despite optimized anticoagulant therapy. We did a comprehensive review of literature analyzing the possible role of immunomodulation in primary APS to provide a broad overview of potentially safe and effective target treatments for managing this devastating disease.

## 1. Introduction

The antiphospholipid syndrome (APS) is characterized by the development of venous and/or arterial thrombosis and pregnancy morbidity in patients with persistent antiphospholipid antibodies (aPL), including lupus anticoagulant (LA), anticardiolipin antibodies (aCL), and anti-β2 glycoprotein-I (β2GPI) antibodies [1]. According to the revised Sydney (or Sapporo) criteria for the classification of true APS, aPL should be detected on at least two occasions, with testing completed 12 or more weeks apart [2]. The APS may be classified in primary APS (PAPS) and secondary antiphospholipid syndrome (SAPS), i.e., APS occurring in the context of another medical condition [1]. Indeed, the presence of aPL may be associated with other conditions such as autoimmune diseases, mainly systemic lupus erythematosus (SLE) [3], occasionally infections [4], drugs [3], and malignancies [5]. Prevalence of the aPL in the general population ranges between 1 and 5% [1], whereas in autoimmune disease, particularly SLE, the prevalence is as high as 30% [3].

APS clinical manifestations could vary from asymptomatic carriers to life-threatening forms characterized by the rapid development of microthrombosis leading to rapid multiple organ failure [1]. Hallmark features of APS are venous and/or arterial thromboembolic events (TEs) and pregnancy morbidity [6]. TEs result in vascular occlusions; the most common types of venous thrombosis include deep vein thrombosis and pulmonary embolism, while strokes and transient ischemic attacks are the most common arterial thrombosis [6,7]. Non-thrombotic events include thrombocytopenia, hemolytic anemia, arthralgia, arthritis, cardiac valve disease, nephropathy, livedo reticularis, migraine, chorea, and epilepsy [1]. These events are also known as “non-criteria manifestations”. According to The Euro-Phospholipid Project [8], which described the baseline characteristics of a cohort of 1000 patients with APS, early fetal losses (<10 weeks) seem to be more frequent (35.4%) than late fetal losses (≥10 weeks) (16.9%). Obstetric manifestations described in APS are pre-eclampsia, eclampsia, and abruptio placentae [4]. 

Catastrophic antiphospholipid syndrome (CAPS) is a life-threatening form of APS occurring in about 1% of cases [9]. It manifests as microangiopathy and affects small vessels of multiple organs, resulting in organ failure [10]. It has been hypothesized that CAPS may be due to the development of systemic inflammatory response syndrome (SIRS), probably due to excessive cytokine release from injured tissues [8]. Infection, surgery, pregnancy, and puerperium are identified triggers of CAPS [11,12]. CAPS-first line therapies include the association of anticoagulation, glucocorticoids, plasma exchange, and/or intravenous immunoglobulins (IVIGs) [13].

Treatment strategies for the management of APS can be differentiated in primary thrombo-prophylaxis in patients with persistent aPL with no prior TEs, and secondary thrombo-prophylaxis, in patients with previous TEs [14,15]. Regarding primary thrombo-prophylaxis, it is essential to underline that not all individuals with high levels of aPL develop thrombosis [15]. Factors that should be taken into account when assessing the risk of TEs are the concomitance of other risk factors for thrombosis and for cardiovascular diseases, the presence of other autoimmune conditions, and the multiple positivity of aPL [15]. Indeed, Pengo et al. reported that multiple positivity of aPL is more frequently associated with TEs than a single test positivity. The triple-positive population of LA, aCL, and anti-β2GPI has been regarded as the highest risk group for TEs and poor neonatal outcomes [16,17]. The authors also showed that in their population TEs were more frequent among male subjects [17]. Kelchtermans et al. [18] reviewed extensive literature to identify the most important criteria for identifying patients at risk for thrombosis. They found more significant correlations with thrombosis for the IgG as compared to the IgM isotype, even though, in a minority of studies, significant associations with thrombosis were found for IgM but not IgG antibodies. Moreover, thrombosis risk is increased with LA or anti-β2GPI occurring alone compared to aCL alone [6,18]. 

However, current therapeutic options for the treatment of APS remain long term anticoagulation with vitamin K antagonist [Warfarin at INR (international normalized ratio) 2–3 in patients with a first venous TE; INR >3.0 or combined antiaggregant-anticoagulant (INR 2.0–3.0) therapy in patients with arterial thrombosis] [19]. Direct oral anticoagulants, such as rivaroxaban, are alternative anticoagulants in patients with APS and venous TEs when a poor control is obtained with standard therapy or in the presence of drug allergy [20]. The currently accepted treatment for APS management during pregnancy is low-dose aspirin plus prophylactic unfractionated or low-molecular-weight heparin (LMWH) [21,22,23,24,25]. Low-dose aspirin (50–100 mg daily) can be administrated as standard primary prevention of fetal loss, administered from the beginning of pregnancy until just before delivery [1]. This may be combined with daily subcutaneous heparin in the face of previous fetal losses using aspirin [26]. Despite the therapy, in approximately 20% of APS cases, a live birth cannot be achieved [27,28,29]. Focusing on the activation of immune cells, to whom it has been recently attributed the role of “second hit” in the APS-associated clotting, could be useful to elucidate the different clinical aspects of the syndrome and debate about the use of immunosuppressive treatments.

## 2. Etiopathogenesis

The etiopathogenesis of APS and many other autoimmune diseases are still not completely understood, and many open questions persist. It is commonly accepted the hypothesis that both environmental factors and genetic predisposition are the underlying mechanisms for the production of aPL [30]. The detection of circulating aPL is an indispensable pre-condition, and aPL are diagnostic markers for and pathogenic drivers of APS. Although aPL are similar to other autoantibodies, they have been linked to several clinical manifestations of the disease. They display different binding specificities and can activate platelets, endothelial cells, monocytes, and several other cells. They also act on plasma coagulation and complement system, two of the key players involved in coagulation and inflammation [31].

Although the aPL family consists of a various group of autoantibodies [31] which interact with a large extent of phospholipid (PL)-binding plasma proteins, a central role in the pathophysiology of APS is played by antibodies against β2GPI [32] directed against a hidden epitope of the β2GPI molecule. β2GPI is a plasma protein circulating in the peripheral blood in a circular form and characterized by five complement control protein (CCP) repeat domains. After binding the fifth domain to negatively charged surfaces, the β2GPI molecule opens up and displays the feature of a fishhook, thus exposing the hidden epitope in the first CCP domain [33,34].

It has been largely demonstrated in vitro that aPL are involved in activating several cells, including endothelial cells, platelets, and (syncitio-) trophoblast cells. The main candidate receptors for β2GPI on cell membranes are Toll-like receptors (TLRs), apolipoprotein E receptor 2, and several cellular receptors [35]. By investigating the role of immune cells in recent years, significant developments in understanding the pathogenetic mechanisms of APS have been achieved. New potential therapeutic interventions, including anti-inflammatory, immunomodulatory, and novel therapeutic windows in the management of the disease, have been explored. In Figure 1 we have summarized the main pathogenetic pathways of the antiphospholipid syndrome and the possible molecular targets of the therapy.

### 2.1. Dendritic Cells in APS

Dendritic cells (DCs) represent the classical link between innate and adaptive immunity, regulating both immunity and tolerance and acting as sentinel cells for our body. They capture, process, and present antigens to T cells, travel via the lymph to specific T cell areas in lymphoid tissues where the interaction between the antigen-loaded MHC complex, and an antigen-specific T cell receptor takes place [30]. 

The importance of DCs in the development of systemic antibody-mediated diseases is illustrated in experimental models, demonstrating that DC activation A20-mediated is a crucial checkpoint in the development of systemic autoimmune diseases. The spontaneous proliferation of conventional and double-negative T cells, conversion to effector cells producing interferon-γ (IFN-γ), and expansion of plasma cells is observed in DCs of mice lacking the ubiquitin editing enzyme A20 (Tnfaip3). Increased prosurvival signals and upregulation of antiapoptotic proteins reduce A20-deficient DCs apoptosis. In addition, this knock-out induces spontaneous maturation and hyper-responsiveness to DCs activation stimuli, inducing self-reactive effector lymphocytes [35].

Furthermore, a fundamental role for DCs in aPL formation has also been proposed; mice that have pinocytosed β2GPI or phagocytosed apoptotic thymocytes start to produce anti-β2GPI antibodies following the injection of bone marrow-derived DCs [36]. 

Apoptotic cells are characterized by a negative charge on their surface. β2GPI easily binds these anionic surfaces acting as an opsonin for apoptotic cells. Human moDCs showed that the binding of β2GPI to an anionic surface strongly enhances the presentation of apoptotic cell-derived epitopes to helper T cells in the context of MHC-II molecules and the capacity of moDCs to stimulate β2GPI-reactive T cells [37,38]. 

In APS patients, compared to patients with other autoimmune diseases (aPL positive or negative), an increased oxidized β2GPI in blood has been described. This condition can induce maturation and activation of human monocyte-derived dendritic cells (MoDCs) thus causing an increased secretion of pro-inflammatory cytokines and the expression of co-stimulatory molecules. Indeed, the pro-inflammatory state of DCs is characterized by an increased cytokine production (IL-1β, IL-10, and tumor necrosis factor (TNFα)), which is observed during APS [39].

Torres-Aguilar and colleagues demonstrated in 2012 that APS tolerogenic DCs, a subtype of MoDCs, can downregulate T cell responses. These DCs significantly decreased the proliferative potential and cytokine production of the autoreactive T cells when β2GPI-loaded tolerogenic DCs are co-cultured with β2GPI-reactive T cells from APS patients [40]. These observations suggest that manipulating DCs to become tolerogenic could be fundamental in downregulating the initiation of an immune response toward β2GPI.

Plasmacytoid DCs (pDCs) display an important role in the pathogenesis of systemic autoimmune diseases even if their specific function in APS has not been completely elucidated. pDCs induce an antiviral immune response by producing large amounts of IFNα and β [41]. It has been demonstrated that aPL display in vitro an activating effect on pDCs by upregulating their TLR7 expression. TLR7 upregulation strongly enhances the secretion of pro-inflammatory cytokines such as IL-1β, TNFα, and IFNα [42].

Type I IFNs augment T and B cell responses, and several autoimmune diseases are characterized by the overexpression of a set of genes known as downstream events after type I IFN stimulation. The hypothesis that pDCs play a key role in the pathogenesis of APS also derives from the evidence that agonists of TLR7 potentiate the induction of APS in mice. 

In addition, hydroxychloroquine (HCQ), a drug frequently used in the treatment of SLE, is involved in the downregulation of TLR signaling, causing a reduction of IFNα production by pDCs in SLE patients [30].

### 2.2. T Cells in APS

Qualitative lymphocyte perturbations are described during APS, raising the possibility that the flow cytometry analysis of the lymphocyte subset could be helpful. Autoreactive β2GPIspecific T-helper (Th)-cells uniformly react with an epitope located in the fifth domain of β2GPI. Several studies reported that stimulation of β2GPI autoreactive T cells results in the predominant secretion of Th1 cytokines like IFNγ. In addition, these T cell autoreactive clones, in turn, stimulate B cells to produce anti-β2GPI antibodies. An increase in Th2 and Th17 cells, and a decrease in Th1 and Tregs, is observed in APS patients [43]. In 2013, Dal Ben and colleagues described a significant decrease in the percentage of peripheral blood Treg cells in primary APS patients as compared to healthy controls [44].

### 2.3. B Cells in APS

Although APS is considered an antibody-mediated disease that may share some similarities with other autoimmune diseases, unique features characterize this condition. A central role is expected for B cells in the pathophysiology of the disease, and a dysregulation in B cell differentiation and B cell tolerance checkpoints is strongly hypothesized; however, only a few small-sized studies addressing B cell homeostasis in APS are available [45]. 

A deregulation at any of one of the three main checkpoints involved in the clearance of the autoreactive B cells can lead to both the altered naïve B cell repertoire (BCR) and the production of autoantibody-producing B cells.

The B cell-stimulating molecules, BAFF (B cell-activating factor, also known as soluble B lymphocyte stimulator (BLyS) and TALL-1) and APRIL (a proliferation-inducing ligand), are closely related ligands of the TNF family [46,47]. The BLyS axis is implicated in the autoimmune diathesis playing a critical role in self-Ag-driven autoimmune B cell activation that occurs in several human autoimmune disorders such as Sjögren’s syndrome, rheumatoid arthritis, IgA nephropathy, and SLE and malignancies [48].

The precise role of this cytokine in the pathogenesis of APS has yet to be elucidated. Experiments on the APS prone mouse model ((NZW × BXSB)F1 mice) showed that BAFF blockade is able to reduce the occurrence of myocardial infarction [49]; nevertheless, no significant effect was obtained on aPL titers in a 12-month follow-up study carried out on patients treated with rituximab [50]. Changes in circulating B cell subsets in primary APS patients have not been demonstrated, and the higher number and variety of autoantibodies observed in patients affected by SLE in comparison with APS could be explained by the differences observed in the naïve B cell repertoire [45].

### 2.4. Monocytes in APS

Compared with other immune cells, monocytes have been extensively studied in APS. Several complications of APS, such as thrombotic and non-thrombotic manifestations, seem to be related to monocyte activation. Monocytes obtained from patients affected by APS are in an activated state characterized by the upregulation of several intracellular pathways, i.e., NFκB, MEK-1/ERK, and p38 MAP kinase [51]. The binding of β2GPI/anti-β2GPI antibody complexes to cellular receptors has been associated with monocyte activation [52].

TLR8 and its co-receptor molecule CD14 are up-regulated in PBMCs from APS patients as compared with those from healthy controls or patients affected by other autoimmune diseases, and TNFα, IL1β, and caspase-1 production is increased by pre-incubation of monocytes with monoclonal aPL. In addition, other TLRs, such as TLR1, TLR2, TLR6, and TLR8 have been related to the monocyte activation status observed in APS. The procoagulant status of monocytes from APS patients is demonstrated by the increased production, surface expression and activity of one of the most important factors involved in coagulation: the tissue factor (TF) via the pro-inflammatory sensitization to TLR 7/8 [30].

Using RT-PCR, it has been shown that exposing monocytes from healthy controls and monocytes from APS patients to low concentrations of lipopolysaccharide (LPS), APS monocytes induced increased levels of TLR2, IL-23, CCL2, CXCL10, IL-1β, and IL-6, while LPS in healthy cells led to IL-6 and STAT3 elevation mRNAs [53].

Monocyte activation is also mediated via TLR4 by Annexin A2, which is considered a potential docking station for β2GP/anti-β2GP antibody complexes [52]. Subsequent signaling induced by TLR4 activation leads to phosphorylation of MyD88, TRIF, IRAK, and of the NF-κB-pathway, which results, among other effects, in TNFα secretion and TF expression.

Monocytes from thrombotic APS patients produce more Annexin A2 than non-thrombotic APS patients and healthy controls, and in line with this consideration, blockage of Annexin A2 reduces the expression of TF on healthy monocytes [51]. 

It has also been observed that monocytes from APS patients overexpress the vascular endothelial growth factor (VEGF) and the VEGF receptor Flt. The blockade of the signaling pathways for p38, ERK-1 MAP-kinase and/or NFκB reduces the expression of both TF and VEGF [30].

The procoagulant state of monocytes in patients with thrombotic APS is more pronounced than in women with obstetric APS. Several studies showed that the circulating levels of aPL modulate TF expression [54]. Two microRNAs (miRNAs) (miR-19b and miR-20a) are downregulated in monocytes from APS patients. In vitro upregulation of these two miRNAs reduces TF expression suggesting that these miRNAs are important in inducing the procoagulant phenotype observed in APS monocytes [55].

The pro-inflammatory status of monocytes is also demonstrated by the increased production of cytokines and reactive oxygen species (ROS). Several drugs interfere with the increased activation state of APS monocytes: chloroquine, CoEnzyme Q10, and a combination of vitamin E and vitamin C or Ubiquinol can modulate monocytes. Recently, losmapimod application in patients with myocardial infarction makes this drug of possible use in APS [30].

### 2.5. Neutrophils in APS

Neutrophils, short lived and terminally differentiated cells, are the most abundant circulating leukocyte and contribute to autoimmune dysregulation in APS, with a phenotype that differs from that observed in healthy subjects [56]. Among peripheral blood mononuclear cells (PBMC), a subset of low-density neutrophils with different properties from normal density granulocytes can also be found. Two types of low-density neutrophils have been described, and currently, there is no consensus regarding whether these cells belong or not to the same cell type. The low-density granulocytes (LDGs) are mainly related to the pro-inflammatory phenotype; indeed, the neutrophil-like myeloid-derived suppressor cells (PMN-MDSC) show an anti-inflammatory phenotype [57]. MDSCs described for the first time in cancer are identified as myeloid progenitor cells with suppressive effects on T cells [58]. However, their function in humans in the context of APS needs further attention to better define their contribution. 

Phagocytosis, oxidative burst, and release of neutrophil extracellular traps (NETs) are all altered in neutrophils APS [59], causing irregular clearance of cell remnants and nuclear material, impairment of ROS production, and antigenic burden. Neutrophil deregulation is also responsible via NETs of the pro-thrombotic pattern involved in cardiovascular events. 

Moreover, neutrophils interact via direct cell-to-cell contact with other immune cells, especially platelets [57]. Neutrophil-platelet interactions play an important role in thrombo-inflammation, and it has been hypothesized that they are also involved in the cardiovascular manifestations observed in APS. LGDs and platelets can release mitochondria and mitochondrial DNA (mtDNA), which are strongly involved in inducing pro-inflammatory cytokines, including type I IFN, IFNγ, IL-6, IL-8, and TNFα, all actors in the context of the autoimmune pathogenesis of APS [56]. 

To drive neutrophil migration, P-selectin glycoprotein ligand 1 (PSGL-1) needs binding P-selectin on activated platelets [60]. PSGL-1 is upregulated in APS, and its blockade results in decreased neutrophil-platelet interactions and protection against thrombo-inflammatory injury. Neutrophil chemotaxis is also mediated by pro-inflammatory cytokines and chemokines released by activated platelets [56]. 

### 2.6. Complement in APS

Over 50 plasma proteins involved in host defense compose the complement system, an integral part of the innate immune system, whose pathways converge at the level of complement component C3, inducing the generation of C5a and the membrane attack complex. Many regulatory mechanisms, including membrane-bound and soluble regulators (like factor H and factor I), constitute this system. Complement and coagulation pathways are closely linked, and anti-β2GPI antibodies are associated with complement activation [61].

It has been suggested that complement is involved in vascular APS following the observation of increased plasma levels of activation products and reduced C3 and C4 levels or CH50 activity in some patients. However, several studies reported contrasting data on the concentrations of C3, C4, and of CH50 activity in patients with APS compared with healthy controls and patients with non-SLE connective tissue diseases such as systemic sclerosis or Sjögren’s syndrome [62]. In animal models, complement blockage has been shown to be critical in protecting animals from both aPL-mediated clotting and fetal loss [38]. 

TF-dependent procoagulant activity is modulated by neutrophil activation of C5a [63], which may lead to fibrinolysis inhibition. Complement activity can also induce the activation of endothelial cells, the expression of adhesion molecules, and an increased procoagulant activity [64]. Placental inflammation and injury are both considered a hallmark of fetal loss in APS and complement activation seems to be strongly involved in this complication [65]. β2GPI is considered as the main complement regulator. Indeed β2GPI displays a structural similarity in factors H and I, making them able to bind complement components C3 and C3b. After the binding of C3 or C3b to membrane bound β2GPI, these molecules undergo a conformational change that enables them to bind factor H, followed by their degradation by factor I. This complement regulatory function of β2GPI was identified in domain I, which contains the binding site of different anti-β2GPI antibodies [66]. Therefore, hypocomplementemia, which suggests complement consumption, could be related to the effects of anti-β2GPI antibodies. It is feasible that these antibodies could trigger complement system although, the effects on APS clinical manifestations in humans require further investigations. 

It has also been suggested that immune complexes in APS bind C1q and activate the classical complement pathway [67]. Antibodies against C1q have been detected in patients with SLE and correlate with lupus nephritis [68]. Moreover, Oku and colleagues reported that anti-C1q antibodies were increased in patients with primary APS (36%) as compared with controls affected by other non-SLE autoimmune disorders [69]. Both animal models of thrombotic APS and clinical evidences, demonstrating higher serum levels of C5b-9 in patients with aPL and stroke, support a role for complement in aPL mediated thrombosis [70]. Others have reported hypocomplementemia and higher levels of complement fragments Bb and C3a in patients with APS; however, the association with APS-related thrombotic events or serologic characteristics is inconsistent [62]. 

## 3. The Current Role of Immunosuppressant Drugs in Antiphospholipid Syndrome

Immunosuppressant drugs, including biological drugs, are widely used to treat several autoimmune conditions, such as rheumatoid arthritis, SLE, Sjögren’s syndrome, and vasculitis. Their safety and effectiveness in these conditions have been largely demonstrated. Similar evidences in the treatment of APS are limited to case reports or case series, and studies on a large scale are currently lacking. According to recent guidelines, immunotherapy is generally not recommended in APS unless required to manage the underlying autoimmune condition or treat the catastrophic variant [1,19]. The suppression of the aPL title through immunotherapy or plasmapheresis is only temporary, and it is not the main aim of the therapy [1] since the role of the aPL level reduction on APS clinical manifestations is not entirely understood. Several authors described a temporary link between aPL reduction and symptoms improvement [10,71,72]; on the other hand, the hypothesis that the two events are causally associated has not been strictly demonstrated. However, immunomodulatory drugs may be an emerging tool in managing such particular situations. Reproductive health is a critical issue whose importance has been addressed by several recent publications [73]. For example, concerning pregnancy management in women with APS, the main purpose of the treatment is to improve both maternal and fetal–newborn health outcomes [14]. Immunosuppressant treatment could be added to conventional therapy in pregnant women with recurrent fetal losses despite treatment with heparin and low dose aspirin. Indeed, in PROMISSE, a prospective observational study of 724 patients, 44% of pregnancies in women with APS and LA resulted in adverse pregnancy outcomes despite conventional therapies [74]. In addition, immunosuppressive therapy may be useful to treat non-thrombotic complications [50,75]. 

We reviewed extensively the relevant literature, analyzing the possible role of immunomodulation in primary APS. Table 1 shows the main treatment strategies, including immunosuppressant drugs, reported in the literature up to the present, and their relative indications are summarized.

### 3.1. Hydroxychloroquine

HCQ is an anti-malarial with immune-modulating, anti-inflammatory, and antithrombotic effects [106]. Indeed, besides its inflammatory action due to the inhibition of cytosolic phospholipase A, and the reduction of the cytokines IL-6 and TNF-α [107], HCQ also inhibits platelet aggregation and arachidonic acid release [20]. Its antithrombotic effect has been demonstrated both in vitro and in vivo. In fact, HCQ decreases the attachment of aPL-β2GPI complex to phospholipid bilayers and cells, protects the annexin V anticoagulant shield from disruption by aPL [108], and decreases thrombus size in murine models of APS [109]. In addition, HCQ reverses the binding of aPL to human placental syncytiotrophoblasts [108] and significantly reduces soluble TF levels in patients with aPL [110]. Wu et al. [111] performed a double-blind study of 134 patients to compare the effectiveness of HCQ and heparin in the prophylaxis of deep venous thrombosis. They detected deep venous thrombosis in six patients in the placebo group, in one patient in the HCQ group, and none in the heparin group, confirming that both heparin and HCQ can diminish thrombosis incidence. 

On the other hand, Johansson et al. [77] evaluated HCQ efficacy as thrombo-prophylaxis after hip surgery, and they found that 2 out of 18 patients treated with HCQ developed deep venous thrombosis compared with 11 out of 17 patients in the control group. Platelet aggregability was inhibited in vitro by high concentrations of HCQ, but these concentrations were not obtained in vivo, and platelet aggregation was not limited in patients treated with HCQ. In a prospective non-randomized study by Schmidt-Tanguy et al. [76], 20 patients with PAPS were treated with HCQ (400 mg daily) associated with oral anticoagulants, while controls were only treated with oral anticoagulants as secondary venous thrombosis prophylaxis. After a three-year follow-up, no recurrent thrombotic events were observed in APS patients receiving oral anticoagulants plus HCQ. In contrast, six recurrent thromboses (30%) were observed in APS patients receiving only oral anticoagulants despite an INR in the therapeutic area. 

Regarding the use of HCQ for the management of APS obstetric complications, anti-β2GP1 antibodies decrease trophoblastic differentiation via TLR4; this effect is re-established by HCQ, suggesting its therapeutic interest in APS pregnancies [112]. A European multicenter retrospective study on 35 pregnancies of 30 patients with APS showed the benefit of HQC addition in patients with refractory obstetrical APS [78]. The authors compared the outcome of pregnancies treated by HCQ with previous pregnancies under the conventional treatment and demonstrated that pregnancy losses decreased from 81% to 19% (*p* < 0.05). In addition, 14 patients with previous refractory obstetrical APS, all with previous pregnancy losses under treatment (aspirin with LMWH in 11 cases and LMWH in 3 cases), the addition of HCQ resulted in increased live-born babies (78%) (*p* < 0.05). Another observational, retrospective, single-center cohort study analyzing 170 pregnancies in 96 women with APS showed that HCQ treatment was associated with a higher rate of live births (67% vs. 57%; *p* = 0.05) and a lower prevalence of aPL-related pregnancy morbidity (47% vs. 63%; *p* = 0.004) [79].

In conclusion, HCQ is routinely not recommended in patients with PAPS [106,113] since large and controlled trials are missing. However, HCQ should be considered in patients with refractory APS symptoms achieving a poor outcome despite conventional therapy [106]. On the other hand, in SLE patients persistently positive for aPL, HCQ led to a reduction in thromboembolic events [76]; thus all SLE patients should be considered for HCQ therapy [19,106]. 

### 3.2. Rituximab

Rituximab is a chimeric monoclonal antibody directed against CD20 [114], approved for non-Hodgkin’s lymphoma, chronic lymphocytic leukemia, microscopic polyangiitis, granulomatosis with polyangiitis, moderate-to-severe pemphigus vulgaris, and refractory rheumatoid arthritis. Moreover, several off-label indications, including SLE, have been shown of some interest [106,115]. 

Rituximab not only reduces aPL levels but also improves clinical manifestation of APS [116]. A well-documented use of rituximab in APS is the management of non-criteria manifestations. A review of literature by Pons et al. [75] analyzed the efficacy of rituximab in 24 patients with non-criteria manifestations of APS, such as thrombocytopenia, skin, neurologic and heart valve involvement, hemolytic anemia, and pulmonary and renal involvement. Eleven patients (45.8%) presented a complete clinical response, seven (29.2%) a partial response, and six (25%) did not respond to rituximab. A 12-month, phase II pilot study by Erkan et al. [50] on adult aPL-positive patients with thrombocytopenia, cardiac valve disease, skin ulcers, aPL nephropathy, and/or cognitive dysfunction, assessed that rituximab was effective in controlling some of these conditions (e.g., skin ulcers and cognitive dysfunction). In this context, one of the most promising applications of rituximab is APS associated thrombocytopenia. aPL plays a role in platelet destruction by binding their membrane [10]. In addition, a close correlation between aPL levels reduction and platelet count increase has been described in a patient with PAPS treated with rituximab for thrombocytopenia refractory to conventional therapies [72].

Similarly, aPL may lead to premature erythrocytes sequestration and destruction, thus causing hemolytic anemia. Some authors observed a rapid and sustained clinical response and a concomitant decrease in aPL levels following rituximab administration in patients with steroid-refractory autoimmune hemolytic anemia associated with APS [87]. In this perspective, rituximab may be considered prior to splenectomy in patients with refractory hemolytic anemia and a high risk of complications following splenectomy. 

APS skin manifestations range from livedo reticularis and livedo racemosa to potentially severe complications such as digital gangrene and extensive cutaneous necrosis. Ulcerations, subungual splinter hemorrhages, superficial venous thrombosis, thrombocytopenic purpura, pseudovasculitic manifestations, and primary anetoderma are also possible [85]. Some case reports showed the efficacy of rituximab combined with plasmapheresis on APS cutaneous involvement [85,86] with the aim to remove aPL by plasma exchange and block the production of new antibodies by depleting B cells [86]. 

The gold standard for the treatment of TEs is indefinite anticoagulation; however, rituximab may be a therapeutic choice in patients experiencing recurrent TEs despite conventional treatment. Indeed, several authors [10,72,89,117,118] observed a dramatic resolution of recurrent and/or severe thrombotic episodes in patients with APS following rituximab introduction. In addition, Veneri et al. [116] reported a case of B cell non-Hodgkin’s lymphoma in which the combination of rituximab with standard chemotherapy led to the complete remission of a severe hypercoagulable state associated with APS.

Rituximab has also been used to treat CAPS with variable response [10,88,89,90,91,119,120]. For example, Rubenstein et al. [89] reported a patient with CAPS resistant to treatment with heparin and methylprednisolone 1 mg/kg/day, successfully treated with rituximab 375 mg/m^2^ once a week for four weeks [89]. A systematic review of the Catastrophic APS Registry (CAPS Registry) [90] identified 20 patients treated with rituximab in association with anticoagulation, corticosteroids, IVIG, plasma exchange, and cyclophosphamide. Fifteen (75%) patients recovered from the acute CAPS episode, one patient did not show symptoms improvement, and four patients (20%) died at the time of the event. 

These evidences, taken together, suggest that rituximab may be an effective alternative treatment in CAPS or APS patients with hematologic and/or microthrombotic manifestations, especially in patients experiencing poor disease control with conventional therapies. 

According to the European League against Rheumatism (EULAR) task force, due to insufficient documentation concerning fetal safety, rituximab should be discontinued before a planned pregnancy [121]. However, so far, there have been nine case reports of patients treated with rituximab during both the first and the second trimester of pregnancy, achieving life birth in all cases [122,123,124,125,126,127,128,129,130]. Six patients were affected by hematological neoplasms, and the remaining had idiopathic thrombocytopenic purpura, vasculitis, and Goodpasture syndrome [122,123,124,125,126,127,128,129,130]. The only adverse effect was a reduction of peripheral B cells in four patients who received rituximab during the second trimester of pregnancy. Indeed, during the second trimester, rituximab is actively transported through the placental by the Fc receptor expressed on the placental surface [131]. The B cell count recovered in six months after birth in all cases. A recent study by Nagata et al. [93] reported the first case of positive pregnancy outcome in a patient with severe thrombocytopenia and refractory obstetric APS following combination therapy with IVIGs and rituximab 600 mg/week for four weeks, from 12th to 15th week of gestation. These evidences suggest that rituximab may be effective and well-tolerated for managing refractory APS during the first trimester; however, larger studies are needed to recommend its use during pregnancy.

### 3.3. Tumor Necrosis Factor-α Blockers

TNF-α is a pivotal mediator of inflammation; hence, TNF-α antagonists can be effective in treating inflammatory disorders in which TNF-α plays a central pathogenic role [84]. TNF-α has been associated with inflammatory mechanisms related to implantation, placentation, and pregnancy evolution [84], and it is involved in aPL-related placental injury and miscarriages [132]. 

Alijotas-Reig et al. [29] reported maternal-fetal outcomes in a series of 18 women with refractory poor aPL-related obstetric outcomes treated with TNF-α blockers during an in vitro fertilization attempt. Patients have already experienced recurrent infertility after triple therapy with LMWH plus aspirin plus HCQ. Sixteen patients were treated with adalimumab and two with certolizumab. Twelve women completed gestation (nine at term and three pre-term) and first-trimester miscarriage or implantation failure occurred in six cases. TNF-α blockers were well tolerated, and no adverse effects or fetal malformations were reported. 

However, almost all TNF-α blockers may show a potential but low embryo–fetal toxicity during pregnancy [84]; for this reason, a careful risk-to-benefit evaluation should be done in pregnant patients. In the pharmacokinetic CrAdLE study, lactating mothers receiving certolizumab were evaluated. The certolizumab pegol concentrations in human breast milk and the average daily infant dose of maternal certolizumab was assessed. The authors stated that no or minimal certolizumab transfer was observed from plasma to breast milk, supporting that certolizumab treatment is compatible with breast feeding [133]. Despite that, certolizumab did not cross the placenta, suggesting a lack of in utero fetal exposure, which indicates no increased rate of major congenital malformations [134]. Hence, certolizumab is an effective and safe treatment during pregnancy and breastfeeding in women with chronic inflammatory diseases.

In addition, biological drugs have been linked with the occurrence of positivity for some autoantibodies and the development of autoimmune diseases, including APS, even if, in most cases, they disappear after the withdrawal of the drug [84,135]. Pérez-De-Lis et al. [135] performed a review of 12,731 cases included in the BIOGEAS Registry, reporting more than 50 different systemic and organ-specific autoimmune disorders in patients exposed to biological therapies. Psoriasis, inflammatory bowel disease, interstitial lung disease, and lupus were the most frequently reported events, and APS occurred in nine patients. A novel study, currently recruiting participants (NCT03152058), aims to evaluate the addition of certolizumab to usual treatments (a heparin agent and low-dose aspirin) in pregnant women with APS. The objective of the study is to determine whether TNF-α blockers may reduce the rate of adverse pregnancy outcomes and alter angiogenic markers of poor placental vascularization.

So far, evidences regarding the possible effectiveness of TNF-α blockers on thrombotic APS are missing.

### 3.4. Eculizumab

Eculizumab is a humanized monoclonal antibody against complement component C5, which inhibits the formation of C5a and C5b, limiting further complement activation [136]. It is currently indicated in paroxysmal nocturnal hemoglobinuria, atypical hemolytic uremic syndrome, and generalized myasthenia gravis [136]. In addition, eculizumab has been considered for the treatment of several immune-mediated diseases, such as lupus nephritis [137], ANCA-associated vasculitis [138], and rheumatoid arthritis [139], with different outcomes. 

Evidences regarding the possible effectiveness of eculizumab on thrombotic APS are scarce. Meroni et al. [95] described the first case of an APS patient with vascular occlusion requiring bypass surgery, successfully treated prophylactically with eculizumab to prevent thrombosis of the bypass.

Eculizumab at the dose of 600–900 mg/week has been used to manage CAPS with significant or complete recovery [9,96,97,98,99,140]. Wig et al. [92] observed a case of CAPS that did not respond to rituximab, subsequently treated with eculizumab with good clinical outcome. Moreover, eculizumab has been used to manage SLE associated refractory CAPS with favorable outcome [141,142].

Further studies have shown that eculizumab may also be useful for primary [101] or secondary prevention [100,101] [NCT01029587] of the development of CAPS in patients undergoing renal transplantation. Indeed, renal transplantation in patients with aPL has been proven historically challenging due to increased risk of thrombosis and allograft failure [101], especially due to the development of thrombotic microangiopathy (TMA), which is often refractory to conventional treatment modalities such as aggressive anticoagulation and plasmapheresis [143]. Eculizumab therapy has been reported to be effective in obtaining the remission of the plasmapheresis-resistant form of TMA related to antiphospholipid syndrome nephropathy (APSN) in patients with SAPS [144,145]. 

Even though data on the safety of eculizumab in pregnancy are scarce, according to Sarno et al. [146], eculizumab may be safe in pregnancy since it is not present in breast milk, and the levels observed in umbilical cord blood samples are not sufficient to affect the concentrations of complement in newborns. However, the use of eculizumab for managing obstetric complications in APS is limited to a single case report by Gustavsen et al. [102]. The authors reported a young primigravida in the second trimester with persistent triple-positive aPL and painful ulcerations of ischemic origin in her right leg. Due to recurrent TEs, she underwent a percutaneous transluminal angioplasty, bypass grafting, and digital amputations. The patient was on lifelong warfarin therapy, but in conjunction with pregnancy, warfarin was substituted with low molecular weight heparin (10,000 IU twice daily) and low dose aspirin (75 mg daily). The patient was considered at high risk of developing CAPS in relation to pregnancy, delivery, and puerperium, based on her multiple previous arterial thromboses and ongoing ischemia during pregnancy. Hence, 600 mg of eculizumab was introduced eight days before delivery, in addition to prophylactic antibiotics, and administrated one day before the delivery again. She developed no postpartum thrombosis, and negligible amounts of eculizumab were detected in the infant. However, although eculizumab treatment can be considered safe in pregnancy, due to these limited data on its effectiveness, the application of eculizumab in APS is currently limited to CAPS and CAPS prevention, especially in patients who are refractory to standard therapies. 

### 3.5. Olendalizumab 

Olendalizumab (ALXN1007) is a humanized monoclonal antibody that can be potentially used in PAPS treatment since it targets the complement inflammatory pathway. As yet, results of Phase IIa Trial of Olendalizumab [NCT02128269], evaluating its safety and tolerability in nine patients with non-criteria manifestations of APS (i.e., aPL-nephropathy, skin ulcers, and thrombocytopenia), have not been fully published. The current results are available at the website www.clincaltrials.gov.

### 3.6. Belimumab

Belimumab is a human immunoglobulin G1λ monoclonal antibody that inhibits BAFF, and it is the only biological agent currently approved for the treatment of non-renal SLE [147]. BAFF blockade in murine studies led to the depletion of B cells, reduced activation of CD4+ cells, decreased expression of adhesion molecules, and reduced deposition of macrophages and DCs [49,106]. These pathophysiological alterations prevent disease onset and prolonging survival in murine APS [49,106]. In patients with PAPS, BAFF levels are increased and correlate with higher adjusted global APS scores [148]. In addition, in SLE patients, belimumab reduces aPL independently from HCQ treatment [149]. In this view, a subset of patients with APS may benefit from BAFF-targeting therapies. Yazici et al. [94] described two primary APS patients treated with belimumab (10 mg/kg) for an aPL-related manifestation. The first patient received belimumab due to recurrent diffuse alveolar hemorrhage and the second one for recurrent skin ulcers. Both patients experienced no new flare, and they were able to taper the prednisone dose. These limited data suggest that belimumab may be a promising target-therapy in refractory APS cases.

### 3.7. Other Immunosuppressants

Other immunosuppressant drugs such as corticosteroids, IVIGs, azathioprine, cyclophosphamide, sirolimus, and defibrotide have been previously used to treat APS patients.

Immunosuppressive doses of corticosteroids (e.g., pulse therapy of methylprednisolone 1000 mg/day for three-five days) have been largely used in combination with anticoagulants in both primary and secondary CAPS patients [150]. The rationale behind their use is the inhibition of the cytokine cascade and of NF-κB, which plays a role in both SIRS and aPL-mediated thrombosis [150,151].

Azathioprine and IVIGs have been used to treat severe thrombocytopenia refractory to prednisone [1,19]. IVIGs may be administrated at the dose of 0.4 g/day/kg for five days in combination with anticoagulation, corticosteroids and/or plasma exchange to treat CAPS patients [150]. They decrease the synthesis of aPL and increase their clearance acting through the Fc receptor; they can also indirectly suppress cytokines and complement system activation and modulate T cell activity [150].

Cyclophosphamide at the dosage of 0.5–1 g/m^2^ is recommended for the treatment of CAPS associated with SLE, but not in PAPS patients [150]. Indeed an analysis of the CAPS registry reported that cyclophosphamide improved SLE-CAPS patient outcome, but it worsened the prognosis of patients with primary CAPS [152].

Chronic vascular lesions due to the activation of the mammalian target of the rapamycin complex (mTORC) through the phosphatidyl-inositol 3-kinase (PI3K)–AKT pathway in patients with the APS has been previously described [103]. Canaud et al. [103] reported 37 patients with APS nephropathy leading to kidney transplantation. Among them, 10 patients treated with the mTOR inhibitor sirolimus showed better graft survival, in line with in vitro studies. Interestingly, this beneficial effect of sirolimus was not observed in aPL-negative patients. Mora-Ramírez et al. [104] described a patient with APS and myocardial infarction requiring percutaneous coronary intervention (PCI) with sirolimus-coated stents, showing a better outcome as compared with paclitaxel-eluting stents. They suggested that “local administration” of sirolimus, but not paclitaxel, might have foreseen the future development of in-stent neointimal hyperplasia and neoangiogenesis. 

A single case report documented the use of defibrotide in CAPS with complete clinical remission [105]. Defibrotide is an adenosine receptor agonist with affinity to adenosine receptors A1 and A2, which could be able to modulate multiple vascular endothelial cell functions, with an antithrombotic effect [153]. Indeed, vascular endothelial cells might be considered an important target of therapy since CAPS pathogenesis may involve concurrent impairment of these cell functions [105]. Although its potential benefit in APS due to its antithrombotic effect, defibrotide is contraindicated in patients receiving systemic anticoagulation or fibrinolytic therapy, and therefore it is not suitable for many APS patients [106].

## 4. Conclusions

Lifelong anticoagulation remains the cornerstone of therapy for thrombotic APS. However, the use of anticoagulation, especially warfarin, may be problematic due to the increased risk of bleeding, particularly in patients with APS non-criteria manifestation (i.e., thrombocytopenia), drug interactions, or comorbidities. Immunosuppressant drugs may be a therapeutic choice in patients with refractory APS symptoms experiencing recurrent TEs despite standard therapy. Innovative therapeutic approaches targeting inflammatory and intracellular signaling pathways involved in aPL mediated pathogenic effects have also been shown to be effective in managing non-criteria manifestations of APS and CAPS, as well as refractory obstetrical complications. Anyhow, the current clinical experience with several immunosuppressant drugs is limited to case reports and case series; for this reason, further randomized controlled clinical trials, together with a better understanding of APS pathogenesis, are needed for identifying novel therapeutic approaches and provide a safe and effective target treatment for potentially devastating diseases.

## Figures and Tables

**Figure 1 biomedicines-09-00132-f001:**
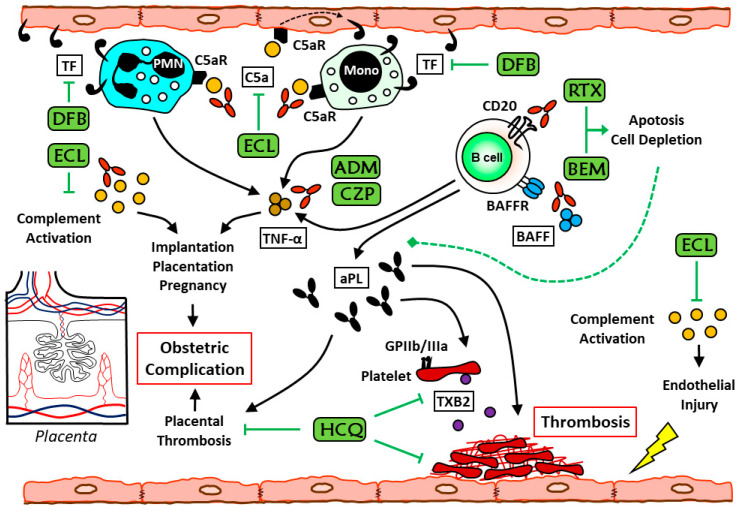
Main pathogenetic pathways in antiphospholipid syndrome (APS) and possible molecular targets of the immunological therapy. ADM: Adalimumab; BEM: belimumab; CZP: certolizumab; ECL: eculizumab; DFB: defibrotide; GPIIb/IIa: Glycoprotein IIb/IIIa; HCQ: hydroxychloroquine; RTX: rituximab; TF: tissue factor; TXB2: Thromboxane B2.

**Table 1 biomedicines-09-00132-t001:** Summary table of studies included in the present systematic literature review, total patients treated (N), clinical indication and outcome.

Drug	Reference	Total Patients Treated (N)	Indication	Outcome Number of Patients = N (%)
HCQ	[76]	20	TEs prevention	No TEs 100% (N = 20)
	[77]	18	TEs prevention	No TEs 88% (N = 16)
	[78]	14	ROC	Live birth 78% (N = 11)
	[79]	31	ROC	Live birth 67% (N = 20)
	[80]	170	TEs and OC prevention	NA
	[81]	1	ROC	Live birth 100% (N = 1)
	[82]	1	ROC	Live birth 100% (N = 1)
	[83]	94	ROC	Live birth 87.2% (N = 82)
Adalimumab	[84]	16	ROC	Live birth 62% (N = 10)
Certolizumab	[84]	2	ROC	Live birth 100% (N = 2)
	NCT03152058	Recruiting	ROC	NA
Rituximab	[50]	19	APS-NCM	Recovery 45% (N = 7)
	[75]	24	APS-NCM	Recovery 45% (N = 11)
	[72]	1	APS-NCM	Recovery 100% (N = 1)
	[71]	1	APS-NCM	Recovery 100% (N = 1)
	[85]	1	APS-NCM	Recovery 100% (N = 1)
	[86]	1	APS-NCM	Recovery 100% (N = 1)
	[87]	1	APS-NCM	Recovery 100% (N = 1)
	[88]	1	APS-NCM	Partial response 100% (N = 1)
	[89]	3	APS-NCM/CAPS	Recovery 100% (N = 3)
	[10]	1	TEs/LNH	No TEs 100% (N = 1)
	[90]	20	CAPS	Recovery 65% (N = 13)
	[91]	1	CAPS	Recovery 65% (N = 13)
	[92]	1	CAPS	No improvement 100% (N = 1)
	[88]	3	CAPS	Recovery 100% (N = 2)
	[93]	1	APS-NCM/ROC	Live birth 100% (N = 1)
Belimumab	[94]	2	APS-NCM	No flares 100% (N = 2)
Eculizumab	[95]	1	TEs prevention	No TEs 100% (N = 1)
	[92]	1	CAPS	Recovery 100% (N = 1)
	[96]	1	CAPS	Recovery 100% (N = 1)
	[9]	1	CAPS	Recovery 100% (N = 1)
	[97]	1	CAPS	Recovery 100% (N = 1)
	[98]	1	CAPS	Recovery 100% (N = 1)
	[99]	1	CAPS	Recovery 100% (N = 1)
	[100]	1	Prevent CAPS after RT	Better graft survival
	[101]	3	Prevent CAPS after RT	Better graft survival
	NCT01029587	1	Prevent CAPS after RT	NA
	[102]	1	ROC/Prevent CAPS	No flares 100% (N = 1)
Olendalizumab	NCT02128269	9	APS-NCM	NA
Sirolimus	[103]	10	APSN and RT	Better graft survival
	[104]	1	PCI in APS	No in-stent stenosis 100% (N = 1)
Defibrotide	[105]	1	CAPS	Recovery 100% (N = 1)

APS-NCM: non-criteria manifestations of APS; APSN: antiphospholipid syndrome nephropathy; CAPS: Catastrophic antiphospholipid syndrome; HCQ: hydroxychloroquine; LNH: non-Hodgkin lymphoma; NA: not available; OC: obstetric complications; PCI: percutaneous coronary intervention; ROC: Refractory obstetric complications; RT: Renal transplantation; TEs: Thromboembolic events.

## Data Availability

No new data were created or analyzed in this study. Data sharing is not applicable to this article.

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
