# Peer review of "Immunosuppressive Treatment in Antiphospholipid Syndrome: Is It Worth It?"

_biomedicines, 2021, doi:10.3390/biomedicines9020132_

Round 1

Reviewer 1 Report

The authors in this review described Immunosuppressive treatment in antiphospholipid syndrome (APS).

In Introduction section they described the essentials of APS, and its clinical manifestations. They also stated the catastrophic APS (CAPS) and treatment strategies for APS.

In Etiopathogenesis section authors stated clearly the important elements in etiopathogenesis of such condition. The Figure 1 is well defined with special focus on main pathogenetic pathways in APS and possible molecular targets for immunological therapy. Further in subsections of this section they described the role of Dendritic cells in APS and the role of T-cells in APS, monocytes in APS, neutrophils in APS and the role of complement in APS. Authors described up to dated findings in these sections with adequate reference.

In further section the role of immunosuppressant drugs in APS were presented. They stated that immunotherapy is generally not recommended in APS unless it is required for the management of the underlying autoimmune condition or to treat the catastrophic variant, with additional role of immunomodulatory drugs. The Table 1 was adequately presented with highlighted important parameters. Additionally, separately they presented the role of hydroxychloroquine in APS with special attention that it should be considered in patients with refractory APS symptoms. Moreover, rituximab role in treatment of APS was described stating that it may be an effective alternative treatment in CAPS or APS patients with hematologic and/or microthrombotic manifestations, especially in patients experiencing poor disease control with conventional therapies. TNF-alfa blockers role in APS were presented with no firm conclusions since evidences regarding the possible effectiveness of TNF-α blockers on thrombotic APS are missing. Also, Eculizumab role in APS was analyzed with recommendations that the application of eculizumab in APS is currently limited to CAPS and prevention of CAPS especially in patients who are refractory to standard therapies. Olendalizumab and Belimumab roles were briefly presented, along with the role of other Immunosuppressants in APS.

The conclusions support the presented data and are presented in clear manner.

This review paper is well organized and the potential benefits of this manuscript are both for improvement in further research and clinical studies for betted understanding of the ethiopathogenetic processes and treatment modes of APS.

Author Response

Point-by-point response to Reviewer #1

We thank the reviewer for his/her comments on the manuscript and we are pleased that he/she believes that this review paper provides potentially useful information for a better understanding of the ethiopathogenetic processes and treatment modes of APS.

The English language and style were checked.

Reviewer 2 Report

The manuscript is interesting and well written. I suggest ot discuss and add as reference paper by Negrini et al. concernin antiphospholipid syndrome.

Author Response

Point-by-point response to Reviewer #2

We thank the reviewer for his/her suggestion.

The reference Clin Exp Med (2017) 17:257–267 has been added in the "Introduction" (page 2) and "section 3" (page 9) and then discussed in "section 3", page 9 as follows: “For example, concerning pregnancy management in women with APS, the main purpose of the treatment is to improve both maternal and fetal–newborn health outcomes (Clin Exp Med (2017) 17:257–267)”.

The English language and style were checked.

Reviewer 3 Report

The article is well written and extremely clear. Authors explored the field in a very exhaustive way, with recent literature and clear explanations.

I think that one limitation is due to the lack of information about treatment and breastfeeding. Your review is on treatment and if you talk about therapy during pregnancy, you should also report some data regardin lactation (e.g CRADLE study for certolizumab). I think it is of major interest.

Another suggestion is to mention the more recent ACR guidelines on pregnanacy and treatment (Sammaritano LR, Bermas BL, Chakravarty EE, Chambers C, Clowse MEB, Lockshin MD, Marder W, Guyatt G, Branch DW, Buyon J, Christopher-Stine L, Crow-Hercher R, Cush J, Druzin M, Kavanaugh A, Laskin CA, Plante L, Salmon J, Simard J, Somers EC, Steen V, Tedeschi SK, Vinet E, White CW, Yazdany J, Barbhaiya M, Bettendorf B, Eudy A, Jayatilleke A, Shah AA, Sullivan N, Tarter LL, Birru Talabi M, Turgunbaev M, Turner A, D'Anci KE. 2020 American College of Rheumatology Guideline for the Management of Reproductive Health in Rheumatic and Musculoskeletal Diseases. Arthritis Rheumatol. 2020 Apr;72(4):529-556. doi: 10.1002/art.41191. Epub 2020 Feb 23. PMID: 32090480).

Author Response

Point-by-point response to Reviewer #3

We thank the reviewer for his/her comments. The manuscript was edited following his/her suggestions.

The reference: Arthritis & Rheumatology Vol. 72, No. 4, April 2020, pp 529–556 was added into the text "section 3" page 9, and a brief statement was also added as follows: "Reproducting health is a critical issue whose importance has been addressed by several recent publications."

The Cradle study was briefly discussed on page 14 in "section 3.3." as follows: "In the pharmacokinetic CrAdLE study, lactating mothers receiving certolizumab were evaluated. The certolizumab pegol concentrations in human breast milk and the average daily infant dose of maternal certolizumab was assessed. The authors stated that no or minimal certolizumab transfer was observed from plasma to breast milk, supporting that certolizumab treatment is compatible with breast feeding"..."Hence, certolizumab is an effective and safe treatment during pregnancy and breastfeeding in women with chronic inflammatory diseases."

The reference was added: Clowse MEB, et al. Ann Rheum Dis 2017;76:1890–1896).

The English language and style were checked.